# Alteration of the corpus callosum in patients with Alzheimer's disease: Deep learning-based assessment

Sadia Kamal[1], Ingyu Park[1], Yeo Jin Kim[2]ⓒ*, Yun Joong Kim[3,4], Unjoo Lee[1]ⓒ*

**1** Department of Electronic Engineering, Hallym University, Chuncheon, Korea, **2** Department of Neurology, Chuncheon Sacred Heart Hospital, Hallym University College of Medicine, Chuncheon, Korea, **3** Department of Neurology, Yonsei University College of Medicine, Seoul, South Korea, **4** Department of Neurology, Yongin Severance Hospital, Yonsei University Health System, Yongin, South Korea

ⓒ These authors contributed equally to this work.
* yjhelena83@gmail.com (YJK); ejlee@hallym.ac.kr (UL)

## Abstract

### Background

Several studies have reported changes in the corpus callosum (CC) in Alzheimer's disease. However, the involved region differed according to the study population and study group. Using deep learning technology, we ensured accurate analysis of the CC in Alzheimer's disease.

### Methods

We used the Open Access Series of Imaging Studies (OASIS) dataset to investigate changes in the CC. The individuals were divided into three groups using the Clinical Dementia Rating (CDR); 94 normal controls (NC) were not demented (NC group, CDR = 0), 56 individuals had very mild dementia (VMD group, CDR = 0.5), and 17 individuals were defined as having mild and moderate dementia (MD group, CDR = 1 or 2). Deep learning technology using a convolutional neural network organized in a U-net architecture was used to segment the CC in the midsagittal plane. Total CC length and regional magnetic resonance imaging (MRI) measurements of the CC were made.

### Results

The total CC length was negatively associated with cognitive function. (beta = -0.139, p = 0.022) Among MRI measurements of the CC, the height of the anterior third (beta = 0.038, p <0.0001) and width of the body (beta = 0.077, p = 0.001) and the height (beta = 0.065, p = 0.001) and area of the splenium (beta = 0.059, p = 0.027) were associated with cognitive function. To distinguish MD from NC and VMD, the receiver operating characteristic analyses of these MRI measurements showed areas under the curves of 0.65–0.74. (total CC length = 0.705, height of the anterior third = 0.735, width of the body = 0.714, height of the splenium = 0.703, area of the splenium = 0.649).

ⓘ OPEN ACCESS

**Data Availability Statement:** We conducted data analysis using OASIS-1 data in https://www.oasis-brains.org/#data. This data is not owned by the authors but we confirm that others would be able to access these data in the same manner as the

authors. The authors did not have any special access privileges that others would not have.

**Funding:** EL was supported by the Basic Science Research Program (2020R1F1A1048281) and Brain Research Program (2017M3A9F1030063) through the National Research Foundation of Korea (NRF) funded by the Ministry of Education. YJK was supported by the NRF grant funded by the Korean government (NRF-2017R1C1B2011637 & NRF-2019R1H1A1035599).

**Competing interests:** The authors have declared that no competing interests exist.

## Conclusions

Among MRI measurements, total CC length, the height of the anterior third and width of the body, and the height and area of the splenium were associated with cognitive decline. They had fair diagnostic validity in distinguishing MD from NC and VMD.

## Introduction

The corpus callosum (CC) is a wide and thick intermediate white matter track that includes several numbers of fibers that connect the two cortical hemispheres and provides interhemispheric transmission of information within the brain [1,2]. Most CC fibers have homotopic connectivity and topographical arrangement [3,4]. Shape changes or decreased CC size are associated with cognitive decline [5,6]. Two mechanisms of callosal atrophy have been proposed: Wallerian degeneration of axons in the white matter due to cortical cell death, and direct myelin breakdown and axonal damage of callosal fibers [7].

Many studies have reported callosal atrophy in patients with AD. Although AD is characterized by medial temporal atrophy, there are changes not only in the gray matter cortical area but also in the white matter tract. The CC is a structure composed only of white matter tract, we assumed that the CC would reflect the white matter change seen in AD. For this reason, CC was studies as a diagnostic marker for MCI and AD in previous studies. However, the regions involved in CC differ between studies. While some studies reported changes in the anterior part of the corpus callosum [8,9], other studies reported splenium atrophy [10,11]. This might be owing to the difference in the study population; however, it might also be due to the methods of segmenting the CC and the differences in defining the subregions [12].

Recently, advances in deep learning technology have made it possible to distinguish MRI images more accurately. Deep learning methods have an advantage in the segmentation of human brain structures for imaging analysis [13]. Traditional methods of imaging segmentation were intensity-based method, atlas-based method, or surface-based method. These methods were time-consuming and sensitive to noise [14]. Additionally, some traditional methods showed low accuracy due to anatomical variability, requiring handcrafting of the researcher for segmentation to increase the accuracy [15]. Using deep learning, it is possible to obtain more accurate results by performing segmentation using an automatic method. This increases the accuracy of medical imaging analysis. In deep learning methods, convolutional networks, which have been widely used in the past, are limited by the size of the available training sets and the size of the considered networks [16]. The U-net architecture is a method that performs very well in biomedical segmentation applications by supplementing the shortcomings of existing segmentation methods [17]. Therefore, in our study, we segmented the CC using deep learning methods, convolutional neural network organized in a U-net architecture, and then measured the characteristics of the CC, and based on this, we investigated the characteristics of the CC in Alzheimer's disease.

## Materials and methods

### Patients

We used the Open Access Series of Imaging Studies (OASIS) cross-sectional dataset [18]. The OASIS dataset was freely released to the scientific community, consisting of 416 individuals aged 18–96 years, from a larger database of individuals who had participated in MRI studies at

Washington University. Subjects with and without dementia were obtained from the longitudinal pool of the Washington University Alzheimer Disease Research Center (ADRC). Subjects with a primary cause of dementia other than AD, active neurological or psychiatric illness, serious head injury, history of clinically meaningful stroke, and use of psychoactive drugs were excluded. The determination of AD or control status is based solely on clinical methods. Clinical dementia rating (CDR) indicated dementia severity. CDR 0 indicated no dementia, CDR 0.5 indicated very mild dementia, and CDR 1 and 2 indicated mild and moderate dementia, respectively.

Of these, we excluded individuals aged less than 60 years to avoid the confounding effects of aging, which is known to influence CC size [19,20]. The individuals were divided into three groups. Individuals with CDR 0 were normal controls (NC), individuals with CDR 0.5 had very mild dementia (VMD), and individuals with CDR 1 and 2 had mild or moderate dementia (MD).

## Image acquisition

For each participant, T1-weighted magnetization prepared rapid gradient-echo images were acquired using a 1.5-T vision scanner (Siemens, Erlangen, Germany) in a single imaging session with the following imaging parameters: sagittal slice thickness, 1.25 mm; no gap; repetition time, 9.7 ms; echo time, 4.0 ms; flip angle, 10˚; inversion time, 20 ms; delay time, 200 ms; and resolution of $256 \times 256$ pixels (voxel size of 1 mm $\times$ 1 mm X 1.25mm).

## Magnetic resonance imaging (MRI) processing

We used SPM12 for CC normalization and segmentation. All the human brain MRI volumes were normalized to the MNI 152 space using SPM12. MNI normalization in SPM12 brings all the individuals to the same space with a resolution of $79 \times 95 \times 79$ and a voxel size of $2 \times 2 \times 2$ mm. After normalization of all the individuals to the MNI space, segmentation was performed using the standard method of SPM12. SPM12 segments the MRI volumes into different parts, such as white matter (WM), gray matter, intracranial cerebral spinal fluid volume, bone, extracranial soft tissue, and background. However, for further preprocessing, we used only segmented WM. After the segmentation of MRI volumes, the midsagittal plane of the WM segmented volume was further used for extracting the CC region. The WM of the mid-sagittal plane was obtained by Talairach transformation using the alignment derived from the line connected the anterior commissure and the posterior commissure in SPM12. Then, segmented CC was passed to train the U-net model. The CC region was extracted by training with the U-net model. The extraction result was the voxel coordinates of the CC area. Using these coordinate values, each characteristic parameter was obtained with MATLAB. MATLAB-based code was developed to extract the CC region from the midsagittal planes of all individuals. Detailed MATLAB-based code was described in S1 Text.

The U-net consists of 23 convolutional layers in total in a contracting path, an expansive path, and a final layer. The contracting path consists of repeated applications of two $3 \times 3$ convolutions and a $2 \times 2$ max pooling operation with stride 2 for down-sampling. The expansive path consists of repeated applications of two $3 \times 3$ convolutions and a $2 \times 2$ convolution for up-sampling. At the final layer, a $1 \times 1$ convolution was used. It was trained with 16 batch sizes, 0.0001 learning rate for 200 epochs with early stopping, where the accuracy was obtained about 91.83%, where the training, validation, and test data set was splitted into a 70:10:20 ratio.

## Definition of MRI measurements

The length of the CC can be measured easily by extracting the most inferior points of the rostrum and splenium. However, the distance measured by this method was assumed to be

inaccurate. The purpose of this bisector line method is to calculate the length of the CC that includes its curvature [21]. In this method, the most inferior points (rows and columns) of the rostrum and splenium are identified. Then, the midpoint of the distance was assumed to be the center of the most inferior part of the rostrum and splenium. The midpoint served as the source for various numbers of radial chords that originated from the evaluated midpoint, each at a regular interval of 10˚. This was performed using the interpolation method in MATLAB. Then, the length of the CC is measured using the Euclidean method for all the lines that connect the midpoints of these chords inside the CC (Fig 1A).

To perform the regional MRI measurements, first, using all the CC images, the extreme rows and columns were identified. (Length, L1 to L6) Then, using the estimated length (from L1 to L6), the callosal length of CC was divided into five equidistance subregions (Part 1 to Part 5) (Fig 1B). For the first fifth part, which corresponded to the genu; width of the genu (a), the width of the rostrum (c), center width of genu and rostrum (b), and lastly, height between the width of genu and width of the rostrum (d) were measured. For the second, third, and fourth positions that corresponded to the trunk, the height of the trunk was measured. This height was evaluated by measuring the height at the center of the center pixel for each region in the second, third, and fourth regions separately. The width of the body was evaluated by measuring the distance between L2 and L5. (Fig 1B) Then, the mean of all distances extracted in each region was detected. For the last fifth, which corresponded to the splenium, rotatory diameter measurement was performed. For this measurement, the fifth region (L5–L6) was extracted, and then the center was assumed to be the crossing point of the maximal horizontal diameter (L5–L6) and the maximal vertical diameter (mid column of horizontal diameter). Ten chords (diameter, D 1–D 10) were estimated using the interpolation method in MATLAB, the length of each starting from the maximal vertical diameter (= D1), at regular intervals of 10˚ until D 10. Then, using the Euclidean distance method, the length of each chord was estimated. Finally, the mean of all the extracted 10 chords, width, and height of the splenium was evaluated (Fig 1C).

## Statistical analysis

The baseline characteristics were presented as mean values for continuous variables and percentages for categorical variables. Differences between each group were confirmed using analysis of variance (ANOVA) for continuous variables and chi-square tests for categorical variables. Differences in each MRI measurement among the groups were evaluated using ANOVA. The association of mini-mental state examination (MMSE) and each MRI measurement was evaluated using multiple linear regression analyses with MMSE as a determinant and each MRI measurement as an outcome variable after controlling for age, sex, education level, and estimated total intracranial volume (eTIV). Receiver operating curve (ROC) analyses of the MRI measurements were used to determine the optimal cutoff values with the Youden index. ROCs were compared among the measurements of the five MRI measurements in a pairwise manner. Statistical significance was defined as $p < 0.05$. Statistical analyses were conducted using SPSS version 25 software (SPSS Inc., Chicago, IL, USA) and dBSTAT version 5 software.

## Results

### Demographics

The baseline demographics are presented in Table 1. The mean age of the patients in the MD group was the highest. The VMD group had the lowest number of women. There were no significant differences in education levels among the groups. (Table 1).

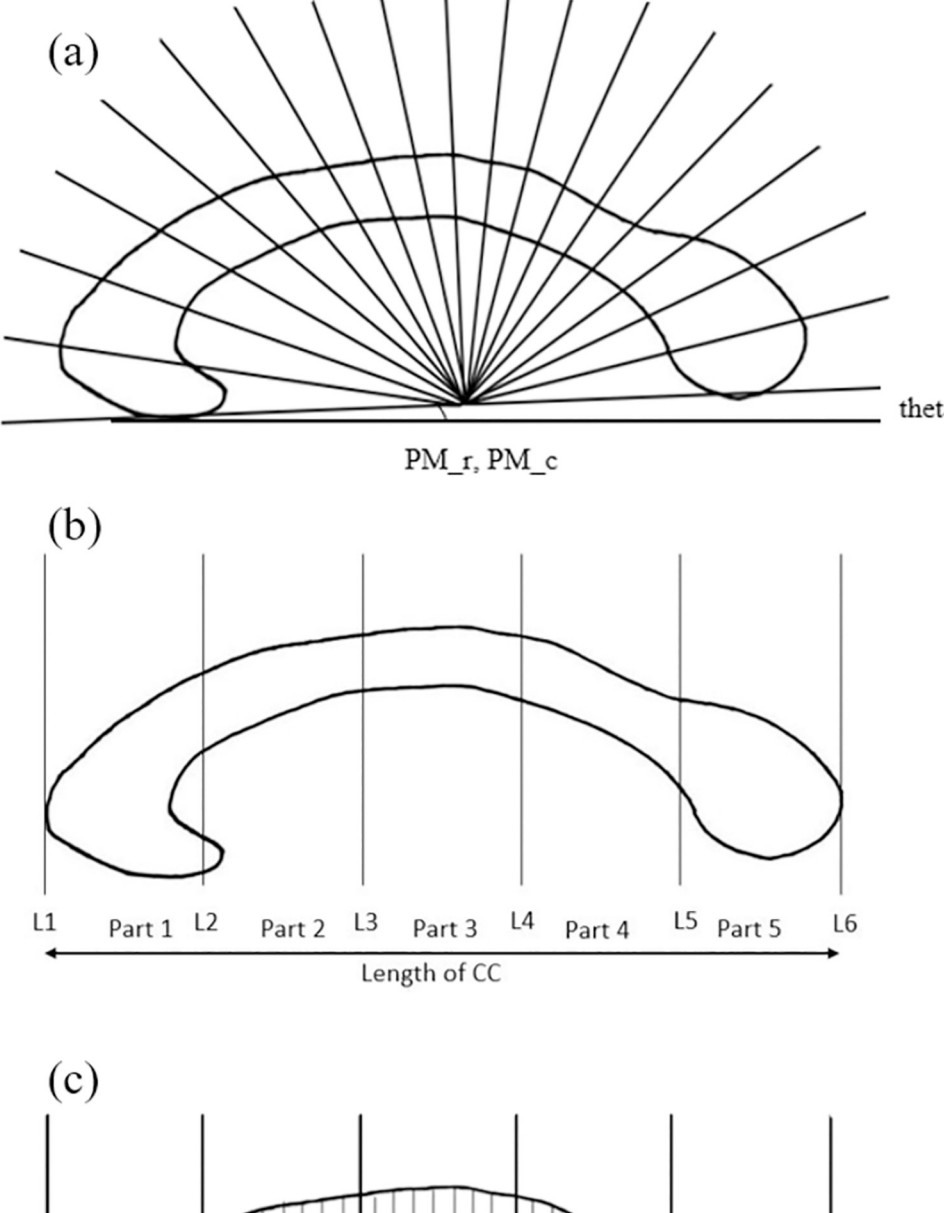

**Fig 1.** Estimation of MRI measurements (a) Evaluation of total CC length (b) Division of CC in five subregions (c) Measurements of each subregion (a, the width of the genu; b, the width of center between genu and rostrum; c, the width of rostrum; and d, height between the width of genu and rostrum; vertical lines of part 2, 3, 4; the height of the anterior, middle, and posterior body portion; lines of part 5, the rotatory diameter of region 5).

**Table 1. Demographic and baseline characteristics.**

|  | NC | VMD | MD | p-value |
|---|---|---|---|---|
| n | 94 | 56 | 17 |  |
| Age (mean±SD) | 73.4±10.66 | 76.9±6.72 | 78.6±7.91 | 0.024 |
| Sex (Female, %) | 72 (76.6) | 30 (53.6) | 13 (76.5) | 0.01 |
| Education (less than college, %) | 32 (34.0) | 26 (46.4) | 10 (58.8) | 0.091 |
| MMSE (mean±SD) | 29.0±1.21 | 25.9±3.11 | 21.2±4.69 | <0.0001 |
| eTIV (mean±SD) | 1447.24±150.962 | 1480.25±169.149 | 1446.65±101.174 | 0.422 |
| MTA, right |  |  |  | <0.0001 |
| 0 | 39 (41.5) | 9 (16.1) | 0 (0) |  |
| 1 | 38 (40.4) | 29 (51.8) | 5 (29.4) |  |
| 2 | 15 (16.0) | 14 (25.0) | 6 (35.3) |  |
| 3 | 2 (2.1) | 4 (7.1) | 6 (35.3) |  |
| 4 | 0 (0) | 0 (0) | 0 (0) |  |
| MTA, left |  |  |  | <0.0001 |
| 0 | 41 (43.6) | 5 (8.9) | 0 (0) |  |
| 1 | 34 (36.2) | 27 (48.2) | 2 (11.8) |  |
| 2 | 16 (17.0) | 15 (26.8) | 8 (47.1) |  |
| 3 | 3 (3.2) | 8 (14.3) | 6 (35.3) |  |
| 4 | 0 (0) | 1 (1.8) | 1 (5.9) |  |

n, number of individuals; NC, normal control; VMD, very mild dementia; MD, mild and moderate dementia; SD, standard deviation; MMSE, Mini-Mental State Examination; eTIV, estimated total intracranial volume; MTA, medial temporal lobe atrophy.

## Comparisons of MRI measurements

There was no significant difference between individuals with normal cognition and individuals with cognitive impairment in all genu or rostrum-related measurements. A significant difference was observed in the height of the anterior third of the body between the NC and MD groups. The MD group also showed a shorter body width than that in the NC group. Both the VMD and MD groups demonstrated a shorter height of the splenium compared to the NC group. Both the VMD and MD groups showed less area of the splenium compared to the NC group. The MD group had a longer total CC length than that in the NC group (Table 2).

## Associations between MMSE and MRI measurements

Among parts of the CC, the body and splenium were associated with the MMSE score, while rostrum and genu did not correlate with the MMSE score. The height of the anterior and middle third of the body were associated with the MMSE score. Body width was also associated with the MMSE score. The height and area of the splenium were associated, while the width of the splenium was not associated with the MMSE score. Total CC length was inversely associated with the MMSE score (Table 3).

When additional analysis was performed by correcting the diagnostic group, only the height of the anterior third of the body and the width of the body were associated with the MMSE score (S1 Table).

## ROC curve analysis of CC measurements for discriminating dementia

ROC analysis of the MRI measurements of the CC to discriminate MD from NC and VMD revealed that the area under the curve (AUC) ranged from 0.65 to 0.74. At the cutoff values with the highest Youden indices, the width of the body showed high sensitivity, while the

**Table 2. Group differences of MRI measurements.**

| | NC | VMD | MD | p-value |
|---|---|---|---|---|
| **Total CC length** | 47.40±2.818* | 48.08±2.890 | 49.68±2.657* | 0.008 |
| **Genu and rostrum** | | | | |
| Width of the genu | 4.20±1.141 | 4.00±1.335 | 3.53±1.328 | 0.105 |
| Width of center between genu and rostrum | 5.03±1.307 | 4.82±1.630 | 4.71±1.105 | 0.534 |
| Width of rostrum | 4.88±2.199 | 4.48±2.587 | 3.88±2.595 | 0.231 |
| Height between the genu and rostrum | 3.19±0.954 | 3.16±1.108 | 3.24±0.970 | 0.962 |
| **Body** | | | | |
| Height of the anterior third | 4.39±0.393* | 4.27±0.436 | 4.05±0.571* | 0.008 |
| Height of the middle third | 3.82±0.421 | 3.71±0.471 | 3.64±0.472 | 0.172 |
| Height of the posterior third | 3.69±0.524 | 3.52±0.514 | 3.47±0.469 | 0.077 |
| Width | 11.90±1.103* | 11.50±1.067 | 11.16±1.327* | 0.014 |
| **Splenium** | | | | |
| Width | 7.88±0.960 | 7.89±1.275 | 7.59±0.939 | 0.555 |
| Height | 5.48±0.936** | 4.91±0.978* | 4.59±0.712* | <0.0001 |
| Area | 6.27±1.313** | 5.53±1.063* | 5.46±1.135* | <0.0001 |

Data presented as means ± SD.

CC, corpus callosum; NC, normal control; VMD, very mild dementia; MD, mild and moderate dementia; SD, standard deviation.

height of the splenium demonstrated the highest specificity. When the ROC analysis results were compared, there was no difference in the AUC among the MRI measurements (Table 4).

## Discussion

Total CC length was negatively associated with cognitive function. Among the parts of the CC, the body and splenium were associated with cognitive function. Among the MRI measurements of the body, the height of the anterior third and total length of the body reflected

**Table 3. Associations between MMSE score and each MRI measurements.**

| | Estimate | SE | R2 | p-value |
|---|---|---|---|---|
| **Total CC length** | -0.139 | 0.06 | 0.242 | 0.022 |
| **Genu and rostrum** | | | | |
| Width of the genu | 0.026 | 0.026 | 0.223 | 0.325 |
| Width of center between genu and rostrum | 0.004 | 0.03 | 0.167 | 0.905 |
| Width of rostrum | 0.077 | 0.055 | 0.059 | 0.161 |
| Height between the genu and rostrum | 0.003 | 0.023 | 0.067 | 0.881 |
| **Body** | | | | |
| Height of the anterior third | 0.038 | 0.009 | 0.266 | <0.0001 |
| Height of the middle third | 0.021 | 0.01 | 0.193 | 0.028 |
| Height of the posterior third | 0.017 | 0.011 | 0.142 | 0.137 |
| Width | 0.077 | 0.023 | 0.273 | 0.001 |
| **Splenium** | | | | |
| Width | 0.016 | 0.025 | 0.022 | 0.519 |
| Height | 0.065 | 0.019 | 0.364 | 0.001 |
| Area | 0.059 | 0.026 | 0.238 | 0.027 |

CC, corpus callosum; MMSE, Mini-Mental State Examination; MRI, magnetic resonance imaging; SE, standard error; R2, coefficient of determination.

**Table 4. Receiver operating curve analysis of corpus callosum measurements for distinguishing dementia from other groups.**

|  | Total CC length | The height of the anterior third of the body | The width of the body | The height of the splenium | The area of the splenium |
|---|---|---|---|---|---|
| AUC | 0.705 | 0.735 | 0.714 | 0.703 | 0.649 |
| SE | 0.059 | 0.073 | 0.063 | 0.059 | 0.071 |
| 95% CI | 0.590–0.820 | 0.592–0.877 | 0.591–0.836 | 0.587–0.819 | 0.511–0.788 |
| Sensitivity | 0.706 | 0.706 | 0.824 | 0.529 | 0.588 |
| Specificity | 0.660 | 0.767 | 0.673 | 0.773 | 0.713 |
| Cutoff value | 48.760 | 4 | 11.250 | 4 | 5.330 |

CC, corpus callosum; AUC, area under the curve; SE, standard error; CI, confidence interval.

cognitive function. Among the measurements of the splenium, height and area were more reflective of cognitive function. These MRI measurements were suitable for distinguishing MD from NC and VMD.

In this study, there were differences in the body and splenium, while there were no differences in MRI measurements between groups for rostrum and genu. The data registered in the OASIS dataset used in this study were obtained from patients diagnosed with AD [18]. Many previous studies have mainly investigated the association between CC atrophy and cognitive impairment in patients with AD [11,22,23]. In this study, both the anterior body part and splenium were affected, which was consistent with previous studies in which AD affected both the anterior and posterior parts of the CC [23–25]. When analyzing the association between MMSE and MRI measurements of the CC as well as comparison by disease group, the body, and splenium of the CC were also associated with MMSE. This study showed that the body and splenium could be markers reflecting the degree of cognitive function, regardless of the disease group. This is in line with a previous study that reported that CC atrophy was associated with global cognitive function even in normal elderly or elderly individuals with only mildly impaired cognitive function [26].

In particular, the size of the splenium in the VMD and MD groups was smaller than that in the normal control group, indicating that the splenium might be a more vulnerable region than other regions of the CC. A previous study also reported that CC atrophy already occurred in patients with mild cognitive impairment, and this study was an extension of the results [12]. Although in some previous studies, atrophy occurred from the anterior to the posterior direction [7,27] other studies reported that the callosal size decreased from the splenium [10,23]. Fibers of splenium connected with parietal, temporal, and occipital cortical regions [28]. One of the possible mechanisms of callosal atrophy is Wallerian degeneration, in which callosal fibers are lost due to the distal loss of the callosal projecting neurons; that is, CC atrophy might be a reflection of cortical neuronal loss [7]. AD is a disease that mainly causes temporal atrophy [29]. In patients with AD, the splenium was more likely to reflect atrophy than the rostrum.

In contrast, in the present study, the total CC length was the longest in the MD group. This was contrary to previous studies and our previous findings that the MD group showed more atrophy. This might indicate that the total CC length reflected less atrophy. In studies analyzing the shape of the CC in other diseases, regional CC thickness reflected the characteristics of the disease better than the total CC length among the measurements representing shape [30,31]. Similarly in our study, the height of the body and splenium reflected cognitive impairment better than the total length. Another possible explanation for why the total CC length was the longest in MD was that the total CC length might reflect distortion. In previous studies, CC circularity was found to be reduced in patients with AD [5]. The total CC length

may reflect reduced circularity. A reduction in circularity could result from a deformation of the CC owing to disease-related enlargement of the lateral ventricles, reflecting an overall atrophic process. In this study, since the length of the CC of the mid-sagittal section was measured, not the volume of the whole CC, irregular changes in the CC due to the deformity caused by the whole-brain atrophy might have increased the total CC length.

Measurements of the body and splenium exhibited fair diagnostic validity for the discrimination of dementia. In particular, the height of the anterior third of the body showed the highest AUC. We also analyzed the value of measurements for discriminating VMD; however, it showed poor diagnostic validity. As for diagnostic methods, our measurements were more meaningful in distinguishing MD, particularly the height of the anterior third of the body, which appears to be useful. Through this study, it was found that changes in the CC were of greater diagnostic value in discriminating patients with mild AD rather than discriminating early in cognitive decline. In previous studies using diffusion tensor imaging, white matter integrity measurements of the CC were reported to have good diagnostic validity for the early detection of MCI and AD [32,33]. However, in previous studies that analyzed CC atrophy, atrophy was evident in AD but not in MCI [24,25]. We used only the morphology of the CC, which might explain why the diagnostic value in the VMD group was not clear, while the diagnostic value in the MD group was more pronounced.

In our study, for accurate measurements of the CC, we used deep learning, U-net, for automatic CC segmentation and extraction. The model was trained using the same features extracted from the developed method as input. The U-net architecture has a particularly good performance in segmentation applications [17]. Therefore, in this study, the accuracy of measurements of the CC might be improved. Although some of the previous research results were different from ours [8,10], we believe that the accuracy of our study would be higher than that of previous studies as we used the deep learning method to perform CC segmentation.

However, this study had several limitations. First, the study individuals were diagnosed clinically, and amyloid pathology was not considered for diagnosis. Second, since this was a cross-sectional study, changes in longitudinal CC atrophy were not considered. Third, since the study population, particularly the MD group, was smaller than the normal group, the possibility that the difference in number between these groups could influence the study results exists. Finally, we used only mid-sagittal plane cross-sectional images for analysis; therefore, we could not measure the total volume. Despite these limitations, we analyzed the characteristics of the CC using a more accurate and automated method using the deep learning method, which enabled us to identify various characteristics of the CC of AD. We anticipate providing new clinical insights into the association between the characteristics of the CC and AD.

## Supporting information

**S1 Table. Associations between MMSE score and each MRI measurements after controlling diagnostic group.**
(DOCX)

**S1 Text. The MATLAB-based code for extracting the corpus callosum.**
(DOCX)

## Acknowledgments

Data was provided by OASIS-1: Cross-Sectional: Principal Investigators: D. Marcus, R, Buckner, J, Csernansky J. Morris; P50 AG05681, P01 AG03991, P01 AG026276, R01 AG021910, P20 MH071616, U24 RR021382.

## Author Contributions

**Conceptualization:** Yeo Jin Kim, Yun Joong Kim, Unjoo Lee.

**Data curation:** Sadia Kamal, Unjoo Lee.

**Formal analysis:** Sadia Kamal, Ingyu Park, Unjoo Lee.

**Funding acquisition:** Unjoo Lee.

**Writing – original draft:** Yeo Jin Kim.

**Writing – review & editing:** Yun Joong Kim, Unjoo Lee.

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
