## [Decision Letter · Decision Letter 0]

21 May 2021

PONE-D-21-14181

Alteration of the corpus callosum in patients with Alzheimer’s disease: Deep learning-based assessment

PLOS ONE

Dear Dr. Kim,

Thank you for submitting your manuscript to PLOS ONE. After careful consideration, we feel that it has merit but does not fully meet PLOS ONE’s publication criteria as it currently stands. Therefore, we invite you to submit a revised version of the manuscript that addresses the points raised during the review process.

Upon my own reading of the manuscript, like Reviewer 1, I had difficulty in understanding exactly how the deep learning method was developed and applied. As this is a key part of the manuscript, many more details need to be provided. Please see the recommendations of Reviewer 1 in this regard. Please also follow the suggestions of Reviewer 2 with respect to assessing the robustness/specificity of your findings.

We look forward to receiving your revised manuscript.

Kind regards,

Niels Bergsland

Academic Editor

PLOS ONE

Journal Requirements:

Reviewers' comments:

Reviewer's Responses to Questions

**Comments to the Author**

1. Is the manuscript technically sound, and do the data support the conclusions?

Reviewer #1: Partly

Reviewer #2: Yes

2. Has the statistical analysis been performed appropriately and rigorously? 

Reviewer #1: Yes

Reviewer #2: Yes

3. Have the authors made all data underlying the findings in their manuscript fully available?

Reviewer #1: Yes

Reviewer #2: Yes

4. Is the manuscript presented in an intelligible fashion and written in standard English?

Reviewer #1: Yes

Reviewer #2: Yes

5. Review Comments to the Author

Reviewer #1: I want to thank the authors for presenting an interesting, and scientifically relevant, manuscript on measuring various aspects of the corpus callosum in patients with Alzheimer’s Disease (VMD and MD) versus normal controls. They present results showing how various features of the corpus callosum (such as total length, subsection height, width, area, and so on) relate to cognitive function. They also perform an ROC analysis for distinguishing the patient populations using the most relevant corpus callosum features. They have a particularly good explanation in the discussion section for why CC length may be increased due to enlargement of ventricles, thus distorting the CC.

Overall, the paper is well-written, with only minor English errors. The results are interesting, but there lacks an explanation of how they performed their deep learning algorithm, which, from my understanding, lays the foundation for their results. This algorithm is not mentioned in any meaningful way in the methods section. This makes the evaluation of the manuscript as a whole difficult.

Major points:

- Introduction:

o 3rd Paragraph on deep learning

This paragraph needs to become clearer as it is too generalizing at the moment. What methods are being compared to the deep learning methods? There are both plenty of conventional programming methods (FreeSurfer, SPM, FSL, and so on), as well as many different types of deep learning methods (CNN, vector-based, PCA, and so on). There’s no need to explain all these in-depth, but it would improve understanding for the reader to know which ones you are referring to, without having to dive deep into the references given.

Not all traditional methods require handcrafting of the researcher (for example, Freesurfer will "recon-all" without user intervention). Although a manual intervention may sometimes improve results.

- Materials and Methods:

o MRI processing section

Did you use SPM12 for CC segmentation? Thus not a deep learning algorithm? It is not clear to me when the deep learning algorithm was applied.

Tell us more about the MATLAB-based code that was used for extracting the CC. This is the main methodology of your research paper, if I understand correctly?

How was the midsagittal slice decided? Or did you extract several slices in the mid-sagittal plane? The manner by which a midsagittal slice is extracted can influence the results quite greatly, as various angles will produce different results for the same CC.

- Discussion

o 6th Paragraph

Here you bring up that you used a Unet for automatic segmentation and extraction. And if I understand correctly, it was trained on the segmentations from SPM12. It is however unclear what the actual accuracy of your Unet algorithm is. Did you perform a cross-validation? This paragraph should be in the methods section, and explain in-depth how you constructed your Unet. How many convolutions? Which optimizer? How many images were used for training and testing? How was it validated? Learning rate? Batch size?

Minor points:

- Abstract:

o Methods section

Please add the abbreviation “OASIS” as this is a well-known dataset.

Please include what type of deep learning technology you used, i.e.: “Deep learning using a convolutional neural network organized in a Unet fashion…”

o Results section

Sometimes you write “MR measurements” and sometimes “MRI measurements”. Both are fine, but I think picking one or the other would improve the flow.

- Introduction:

o 4th Paragraph

Again, it would be nice to know which deep learning method you are using.

If using a Unet, there could be value in adding Ronneberger et al.’s article where the Unet was first presented.

- Materals and Methods:

o Patients section

In the second paragraph you mention that you exclude individuals aged less than 60 years to avoid the confounding of aging, which is known to influence CC size. There is no reference for this. Here’s a reference saying that there is no size difference due to age (which is what makes CC such a promising biomarker for patients with diseases where it actually does atrophy). PMID: 11445261.

o Image acquisition section

A voxel is a three-dimensional pixel. If I understand your dataset correctly, it should state: "1 mm X 1 mm X 1.25 mm".

o MRI processing section

What are you referring to as "soft tissue"? Aren't both white matter and gray matter soft tissues?

- Discussion

o 1st Paragraph

Mixing of past tense and present tense in two adjacent sentences, which I believe are both referring to the results of your paper (i.e. they should both be in past tense).

o Last Paragraph

How was the mid-sagittal slice decided upon and extracted?

Reviewer #2: The present study describes characteristics of the corpus callosum in older individuals with normal cognition (CDR=0), ‘mild dementia’ (CDR=0.5), and dementia (CDR=1-2), assesses associations between corpus callosum characteristics and MMSE score, and the diagnostic discriminative value of corpus callosum characteristics. I do not have the proper background to review the deep learning methods used in the paper, but I have some remarks based on other aspects of the paper:

- It seems that the terms ‘dementia’ and ‘Alzheimer’s disease (AD)’ are used interchangeably throughout the paper. However, not all patients with dementia have AD. Some additional information on the diagnostic background of the cases would be helpful to clarify this. Was the clinical diagnosis for all cases with ‘mild dementia’ and ‘dementia’ Alzheimer’s disease? What diagnostic criteria were used? How was ‘normal cognition’ assessed?

- When analyzing the associations between corpus callosum characteristics and MMSE score, it would be helpful to control for diagnosis (as a dummy variable) to make sure that the associations are not driven by diagnostic groups.

- The relevance of assessing corpus callosum characteristics in Alzheimer’s disease is unclear. Why are the authors interested in this structure? What could be the added value of this marker compared with for example medial temporal lobe atrophy (i.e., what could be the added diagnostic discriminative value of corpus callosum characteristics on top of established measures such as medial temporal lobe atrophy)?

- Similar to the previous remark: The authors find associations between corpus callosum characteristics and cognition, but is this not a reflection of cerebral atrophy in general? Could the authors control for cortical/global atrophy in their analyses? Or perhaps white matter atrophy?

- Is any additional information available on the MRI characteristics of these cases, such as vascular damage / medial temporal lobe atrophy scores? It would be helpful to include this for better characterization of the study sample.

6. PLOS authors have the option to publish the peer review history of their article (what does this mean?). If published, this will include your full peer review and any attached files.

Reviewer #1: **Yes: **Michael Platten

Reviewer #2: **Yes: **Whitney Freeze

---

## [Author Response · Author response to Decision Letter 0]

7 Aug 2021

Manuscript: PONE-D-21-14181

Title: Alteration of the corpus callosum in patients with Alzheimer’s disease: Deep learning-based assessment

Reviewer #1: I want to thank the authors for presenting an interesting, and scientifically relevant, manuscript on measuring various aspects of the corpus callosum in patients with Alzheimer’s Disease (VMD and MD) versus normal controls. They present results showing how various features of the corpus callosum (such as total length, subsection height, width, area, and so on) relate to cognitive function. They also perform an ROC analysis for distinguishing the patient populations using the most relevant corpus callosum features. They have a particularly good explanation in the discussion section for why CC length may be increased due to enlargement of ventricles, thus distorting the CC.

Overall, the paper is well-written, with only minor English errors. The results are interesting, but there lacks an explanation of how they performed their deep learning algorithm, which, from my understanding, lays the foundation for their results. This algorithm is not mentioned in any meaningful way in the methods section. This makes the evaluation of the manuscript as a whole difficult.

Major points:

1.Introduction:

o 3rd Paragraph on deep learning

1-1. This paragraph needs to become clearer as it is too generalizing at the moment. What methods are being compared to the deep learning methods? There are both plenty of conventional programming methods (FreeSurfer, SPM, FSL, and so on), as well as many different types of deep learning methods (CNN, vector-based, PCA, and so on). There’s no need to explain all these in-depth, but it would improve understanding for the reader to know which ones you are referring to, without having to dive deep into the references given.

In response to reviewer’s comment, we added sentences to the third paragraph of introduction section as follows. (Introduction section, Page 5, Line 20 – Page 6, Line 7)

“Traditional methods of imaging segmentation were intensity-based method, atlas-based method, or surface-based method. These methods were time-consuming and sensitive to noise. [14] Additionally, some traditional methods showed low accuracy due to anatomical variability, requiring handcrafting of the researcher for segmentation to increase the accuracy. [15] Using deep learning, it is possible to obtain more accurate results by performing segmentation using an automatic method. This increases the accuracy of medical imaging analysis. In deep learning methods, convolutional networks, which have been widely used in the past, are limited by the size of the available training sets and the size of the considered networks. [16] The u-net architecture is a method that performs very well in biomedical segmentation applications by supplementing the shortcomings of existing segmentation methods. [17]”

1-2. Not all traditional methods require handcrafting of the researcher (for example, Freesurfer will "recon-all" without user intervention). Although a manual intervention may sometimes improve results.

In response to reviewer’s comment, we modified the third paragraph of introduction section as follows. (Introduction section, Page 5, Line 22 – Page 6, Line 2)

“Additionally, some traditional methods showed low accuracy due to anatomical variability, requiring handcrafting of the researcher for segmentation to increase the accuracy. [15] Using deep learning, it is possible to obtain more accurate results by performing segmentation using an automatic method.”

2. Materials and Methods:

o MRI processing section

2-1. Did you use SPM12 for CC segmentation? Thus not a deep learning algorithm? It is not clear to me when the deep learning algorithm was applied.

Response: We used SPM 12 for CC segmentation. Then, segmented CC was passed to train the U-net model. The CC region was extracted by training with U-net model. This content was added to the Materials and Methods section as follow. (Materials and Methods section, Page 8, Line 16 - Line 18)

“Then, segmented CC was passed to train the U-net model. The CC region was extracted by training with the U-net model. The extraction result was the voxel coordinates of the CC area.”

2-2. Tell us more about the MATLAB-based code that was used for extracting the CC. This is the main methodology of your research paper, if I understand correctly?

Response: The MATLAB code for extracting the CC was as follows.

a=spm_vol('c2orig0001.nii'); % real the volume

b=spm_read_vols(a); % read the volume into image matrix

data=squeeze(b(128,:,:)); % extract the sagittal slice; since data is 79x95x79, we will take 39th/40th slice

figure(1),imshow(data,[]); % show the image (extract_midslice.jpg)

bw=bwareaopen(data,100); % remove small objects; objects with pixels less than 100

labelbw=bwlabel(bw); % label the image

figure(3),imshow(bw);% display labeled image (labeled_image.jpg)

figure(4),imshow(labelbw==2); % extract the appropriate label image which has corpus callosum (Segmented_CC.jpg)

We added this code as supplementary text.

2-3. How was the midsagittal slice decided? Or did you extract several slices in the mid-sagittal plane? The manner by which a midsagittal slice is extracted can influence the results quite greatly, as various angles will produce different results for the same CC.

In response to reviewer’s comment, we added the method of mid-sagittal plane decision in Materials and Methods section as follows. (Materials and Methods section, Page 8, Line 22 - Page 9, Line 7)

“The WM of the mid-sagittal plane was obtained by Talairach transformation using the alignment derived from the line connected the anterior commissure and the posterior commissure in SPM12.”

3. Discussion

o 6th Paragraph

Here you bring up that you used a Unet for automatic segmentation and extraction. And if I understand correctly, it was trained on the segmentations from SPM12. It is however unclear what the actual accuracy of your Unet algorithm is. Did you perform a cross-validation? This paragraph should be in the methods section, and explain in-depth how you constructed your Unet. How many convolutions? Which optimizer? How many images were used for training and testing? How was it validated? Learning rate? Batch size?

In response to reviewer’s comment, we added sentences to the Materials and Methods section as follows. (Materials and Methods section, Page 8, Line 22 - Page 9, Line 7)

“The u-net consists of 23 convolutional layers in total in a contracting path, an expansive path, and a final layer. The contracting path consists of repeated applications of two 3 × 3 convolutions and a 2 × 2 max pooling operation with stride 2 for down-sampling. The expansive path consists of repeated applications of two 3 × 3 convolutions and a 2 × 2 convolution for up-sampling. At the final layer, a 1 × 1 convolution was used. It was trained with 16 batch sizes, 0.0001 learning rate for 200 epochs with early stopping, where the accuracy was obtained about 91.83%, where the training, validation, and test data set was splitted into a 70:10:20 ratio.”

Minor points:

1. Abstract:

1-1. Methods section

1-1-1. Please add the abbreviation “OASIS” as this is a well-known dataset.

Response: We added the abbreviation “OASIS”. (Abstract section, Page 3, Line 7)

1-1-2. Please include what type of deep learning technology you used, i.e.: “Deep learning using a convolutional neural network organized in a Unet fashion…”

Response: As your recommendation, we modified this sentence as follows. (Abstract section, Page 3, Line 11 – Line 13)

“Deep learning technology using a convolutional neural network organized in a U-net architecture was used to segment the CC in the midsagittal plane.”

1-2. Results section

Sometimes you write “MR measurements” and sometimes “MRI measurements”. Both are fine, but I think picking one or the other would improve the flow.

Response: In response to reviewer’s comment, we changed MR measurements to MRI measurements.

2. Introduction:

2-1. 4th Paragraph

2-1-1. Again, it would be nice to know which deep learning method you are using.

Response: In response to reviewer’s comment, we modified the fourth paragraph of introduction section as follows. (Introduction section, Page 6, Line 7 – Line 10)

“Therefore, in our study, we segmented the CC using deep learning methods, convolutional neural network organized in a U-net architecture, and then measured the characteristics of the CC, and based on this, we investigated the characteristics of the CC in Alzheimer's disease.”

2-1-2. If using a Unet, there could be value in adding Ronneberger et al.’s article where the Unet was first presented.

Response: In response to reviewer’s comment, we added the Ronneberger et al.’s article as reference number 17.

17. Ronneberger O, Fischer P, Brox T, editors. U-Net: Convolutional Networks for Biomedical Image Segmentation2015; Cham: Springer International Publishing.

2-2. Materals and Methods:

2-2-1. Patients section

In the second paragraph you mention that you exclude individuals aged less than 60 years to avoid the confounding of aging, which is known to influence CC size. There is no reference for this. Here’s a reference saying that there is no size difference due to age (which is what makes CC such a promising biomarker for patients with diseases where it actually does atrophy). PMID: 11445261.

Response: Thank you for your recommendation. However, we found that there were studies that reported corpus callosum size decline in aging. Therefore, we added these studies as references.

22. Hopper KD, Patel S, Cann TS, Wilcox T, Schaeffer JM. The relationship of age, gender, handedness, and sidedness to the size of the corpus callosum. Academic Radiology. 1994;1(3):243-8. doi: 10.1016/s1076-6332(05)80723-8.

23. Janowsky JS, Kaye JA, Carper RA. Atrophy of the corpus callosum in Alzheimer's disease versus healthy aging. J Am Geriatr Soc. 1996;44(7):798-803. Epub 1996/07/01. doi: 10.1111/j.1532-5415.1996.tb03736.x. PubMed PMID: 8675927.

2-2-2. Image acquisition section

A voxel is a three-dimensional pixel. If I understand your dataset correctly, it should state: "1 mm X 1 mm X 1.25 mm".

Response: In response to reviewer’s comment, this issue has been corrected as recommended by reviewer. (Materials and Methods section, Page 8, Line 3)

2-2-3. MRI processing section

What are you referring to as "soft tissue"? Aren't both white matter and gray matter soft tissues?

In this article, soft tissue meant extracranial soft tissue, and in response to the reviewer’s comment, we changed ‘soft tissue’ to ‘extracranial soft tissue’. (Materials and Methods section, Page 8, Line 12)

3. Discussion

3-1. 1st Paragraph

Mixing of past tense and present tense in two adjacent sentences, which I believe are both referring to the results of your paper (i.e. they should both be in past tense).

In response to reviewer’s comment, we changed the sentences to the past tense. 

3-2. Last Paragraph

How was the mid-sagittal slice decided upon and extracted?

In response to reviewer’s comment, we added the method of mid-sagittal plane decision in Materials and Methods section as follows. (Materials and Methods section, Page 8, Line 22 - Page 9, Line 7)

“The WM of the mid-sagittal plane was obtained by Talairach transformation using the alignment derived from the line connected the anterior commissure and the posterior commissure in SPM12.”

Reviewer #2: The present study describes characteristics of the corpus callosum in older individuals with normal cognition (CDR=0), ‘mild dementia’ (CDR=0.5), and dementia (CDR=1-2), assesses associations between corpus callosum characteristics and MMSE score, and the diagnostic discriminative value of corpus callosum characteristics. I do not have the proper background to review the deep learning methods used in the paper, but I have some remarks based on other aspects of the paper:

1. It seems that the terms ‘dementia’ and ‘Alzheimer’s disease (AD)’ are used interchangeably throughout the paper. However, not all patients with dementia have AD. Some additional information on the diagnostic background of the cases would be helpful to clarify this. Was the clinical diagnosis for all cases with ‘mild dementia’ and ‘dementia’ Alzheimer’s disease? What diagnostic criteria were used? How was ‘normal cognition’ assessed?

Response: This study used the OASIS dataset, and the definitions of subjects revealed in the OASIS dataset are as follows.

“Subjects aged 18 to 96 years were selected from a larger database of individuals who had participated in MRI studies at Washington University. Especially, older subjects, aged 60 and older, with and without dementia were obtained from the longitudinal pool of the Washington University Alzheimer Disease Research Center (ADRC). Older adults underwent the ADRC’s full clinical assessment as described below. Subjects with a primary cause of dementia other than AD, active neurological or psychiatric illness, serious head injury, history of clinically meaningful stroke, and use of psychoactive drugs were excluded, as were subjects with gross anatomical abnormalities evident in their MRI images. Dementia status was established and staged using the CDR scale. The determination of AD or control status is based solely on clinical methods, without reference to psychometric performance, and any potential alternative causes of dementia must be absent. The diagnosis of AD is based on clinical information that the subject has experienced gradual onset and progression of decline in memory and other cognitive and functional domains. A global CDR of 0 indicates no dementia, and CDRs of 0.5, 1, 2, and 3 represent very mild, mild, moderate, and severe dementia respectively. These methods allow for the clinical diagnosis of AD in individuals with a CDR of 0.5 or greater, based on standard criteria, that is confirmed by histopathological examination in 93% of the individuals, even for those in the earliest symptomatic stage (CDR 0.5) of AD who elsewhere may be considered to represent ‘mild cognitive impairment”.

Therefore, dementia in our study refers to Alzheimer’s disease.

In response to reviewer’s comment, we added sentences in Patients of Materials and Methods section as follow. (Materials and Methods section, Page 7, Line 6 - Line 11)

“Subjects with and without dementia were obtained from the longitudinal pool of the Washington University Alzheimer Disease Research Center (ADRC). Subjects with a primary cause of dementia other than AD, active neurological or psychiatric illness, serious head injury, history of clinically meaningful stroke, and use of psychoactive drugs were excluded. The determination of AD or control status is based solely on clinical methods.”

2. When analyzing the associations between corpus callosum characteristics and MMSE score, it would be helpful to control for diagnosis (as a dummy variable) to make sure that the associations are not driven by diagnostic groups.

Response: As your recommendation, we further controlled for the diagnostic group in the analysis of the associations between corpus callosum characteristics and MMSE score as below. We added this results as Supplementary table e-1. 

Supplementary table e-1. Associations between MMSE score and each MRI measurements after controlling diagnostic group

 Estimate SE R2 p-value

Total CC length -0.037 0.082 0.260 0.655

Genu and rostrum 

Width of the genu 0.002 0.036 0.229 0.945

Width of center between genu and rostrum 0.001 0.042 0.168 0.977

Width of rostrum 0.060 0.076 0.061 0.427

Height between the genu and rostrum 0.001 0.032 0.069 0.983

Body 

Height of the anterior third 0.046 0.012 0.273 <0.0001

Height of the middle third 0.022 0.013 0.193 0.103

Height of the posterior third 0.015 0.016 0.143 0.331

Width 0.083 0.032 0.273 0.010

Splenium 

Width 0.009 0.035 0.030 0.799

Height 0.031 0.025 0.381 0.222

Area 0.016 0.036 0.263 0.656

CC, corpus callosum; MMSE, Mini-Mental State Examination; MRI, magnetic resonance imaging; SE, standard error; R2, coefficient of determination.

And we also these results added to Results section as follows. (Results section, Page 15, Line 4 - Line 6)

“When additional analysis was performed by correcting the diagnostic group, only the height of the anterior third of the body and the width of the body were associated with the MMSE score. (Supplementary table e-1)”

3. The relevance of assessing corpus callosum characteristics in Alzheimer’s disease is unclear. Why are the authors interested in this structure? What could be the added value of this marker compared with for example medial temporal lobe atrophy (i.e., what could be the added diagnostic discriminative value of corpus callosum characteristics on top of established measures such as medial temporal lobe atrophy)?

Response: Although Alzheimer’s disease is characterized by medial temporal atrophy, it has been known in previous studies that changes in gray matter cortical area as well as white matter tracts are seen. The corpus callosum is a structure composed only of white matter tract, and we investigated the characteristics of the corpus callosum, it would reflect the white matter change seen in AD. We added these to the introduction section as follows. (Introduction section, Page 5, Line 9 - Line 13)

“Although AD is characterized by medial temporal atrophy, there are changes not only in the gray matter cortical area but also in the white matter tract. The CC is a structure composed only of white matter tract, we assumed that the CC would reflect the white matter change seen in AD. For this reason, CC was studies as a diagnostic marker for MCI and AD in previous studies.”

4. Similar to the previous remark: The authors find associations between corpus callosum characteristics and cognition, but is this not a reflection of cerebral atrophy in general? Could the authors control for cortical/global atrophy in their analyses? Or perhaps white matter atrophy?

Response: Our findings are already estimated total intracranial volume (eTIV)-corrected values. Therefore, we think it represents the change of the corpus callosum that is not affected by global atrophy.

5. Is any additional information available on the MRI characteristics of these cases, such as vascular damage / medial temporal lobe atrophy scores? It would be helpful to include this for better characterization of the study sample.

Response: In response to reviewer’s comment, we added the medial temporal lobe atrophy score to Table 1.

Table 1. Demographic and baseline characteristics

 NC VMD MD p-value

n 94 56 17 

Age (mean±SD) 73.4±10.66 76.9±6.72 78.6±7.91 0.024

Sex (Female, %) 72 (76.6) 30 (53.6) 13 (76.5) 0.01

Education (less than college, %) 32 (34.0) 26 (46.4) 10 (58.8) 0.091

MMSE (mean±SD) 29.0±1.21 25.9±3.11 21.2±4.69 <0.0001

eTIV (mean±SD) 1447.24±150.962 1480.25±169.149 1446.65±101.174 0.422

MTA, right <0.0001

0 39 (41.5) 9 (16.1) 0 (0) 

1 38 (40.4) 29 (51.8) 5 (29.4) 

2 15 (16.0) 14 (25.0) 6 (35.3) 

3 2 (2.1) 4 (7.1) 6 (35.3) 

4 0 (0) 0 (0) 0 (0) 

MTA, left <0.0001

0 41 (43.6) 5 (8.9) 0 (0) 

1 34 (36.2) 27 (48.2) 2 (11.8) 

2 16 (17.0) 15 (26.8) 8 (47.1) 

3 3 (3.2) 8 (14.3) 6 (35.3) 

4 0 (0) 1 (1.8) 1 (5.9) 

n, number of individuals; NC, normal control; VMD, very mild dementia; MD, mild and moderate dementia; SD, standard deviation; MMSE, Mini-Mental State Examination; eTIV, estimated total intracranial volume; MTA, medial temporal lobe atrophy

---

## [Decision Letter · Decision Letter 1]

23 Aug 2021

PONE-D-21-14181R1

Alteration of the corpus callosum in patients with Alzheimer’s disease: Deep learning-based assessment

PLOS ONE

Dear Dr. Kim,

Thank you for submitting your manuscript to PLOS ONE. After careful consideration, we feel that it has merit but does not fully meet PLOS ONE’s publication criteria as it currently stands. Therefore, we invite you to submit a revised version of the manuscript that addresses the points raised during the review process.

Please carefully consider the point regarding cross-validation, as suggested by the Reviewer.

We look forward to receiving your revised manuscript.

Kind regards,

Niels Bergsland

Academic Editor

PLOS ONE

Journal Requirements:

Reviewers' comments:

Reviewer's Responses to Questions

**Comments to the Author**

1. If the authors have adequately addressed your comments raised in a previous round of review and you feel that this manuscript is now acceptable for publication, you may indicate that here to bypass the “Comments to the Author” section, enter your conflict of interest statement in the “Confidential to Editor” section, and submit your "Accept" recommendation.

Reviewer #1: All comments have been addressed

Reviewer #2: All comments have been addressed

2. Is the manuscript technically sound, and do the data support the conclusions?

Reviewer #1: Partly

Reviewer #2: Yes

3. Has the statistical analysis been performed appropriately and rigorously? 

Reviewer #1: Yes

Reviewer #2: Yes

4. Have the authors made all data underlying the findings in their manuscript fully available?

Reviewer #1: Yes

Reviewer #2: Yes

5. Is the manuscript presented in an intelligible fashion and written in standard English?

Reviewer #1: Yes

Reviewer #2: Yes

6. Review Comments to the Author

Reviewer #1: Thanks to the authors for their re-working of the manuscript. You have done a good job.

There is just one last thing that I think is necessary before acceptance. You have now presented the "training, validation, and testing" numbers. You have an accuracy of 91% with early termination.

- Firstly, it's unclear if this accuracy is reflecting the validation or testing. I assume testing, but it could be both, as it seemed to have influenced an "early stop" when, I again assume, you reached the highest accuracy level.

There's a bias in exiting training early at the highest accuracy level, as it can represent an overfitting of your own data.

Considering the above stated, in combination with the fact that you only have a small sample size (100 - which is not atypical for biomedical scenarios - and thus why you aptly chose the U-net). I would like to see a cross-validation of your data (I'd recommend at least K-fold: 10, unless you want to do leave-one-out cross-validation). This will give you a better and more true reflection of the performance of your algorithm. It is not uncommon to just do a cross-validation with training and validation data (i.e. skip the test data -> you need as much as you can for the training). Also, have a pre-set number of epochs, and choose the accuracy of the last epoch (i.e. not necessarily the "highest accuracy"). When you do the cross-validation there should not be any tuning of the hyperparameters, as this will affect the results. In the end we are interested in how your algorithm performs without having to tweak it every single time it's applied.

Also, please describe your metric "accuracy". I believe a dice-score metric would be the absolute best for your scenario, but several papers present the classic accuracy (TP + TN / TP + TN + FP + FN), as this tends to show higher accuracy.

Reviewer #2: The authors have addressed all of the comments sufficiently. I have no further comments on the manuscript.

7. PLOS authors have the option to publish the peer review history of their article (what does this mean?). If published, this will include your full peer review and any attached files.

Reviewer #1: **Yes: **Michael Platten

Reviewer #2: **Yes: **Whitney Freeze

---

## [Author Response · Author response to Decision Letter 1]

7 Oct 2021

In response to reviewer’s comment, we performed 10-fold cross-validation as follows. And we used Intersection-Over_Union (IOU) Matric for accuracy (TP/TP+FP+FN)

Plots of accuracies and losses of validation per each k

k Accuracies and losses of validation data Accuracies of test data

1 0.974044

2 0.974546

3 0.974053

4 0.974123

5 0.973244

6 0.974320

7 0.974123

8 0.974245

9 0.974196

10 0.973881

---

## [Decision Letter · Decision Letter 2]

12 Oct 2021

Alteration of the corpus callosum in patients with Alzheimer’s disease: Deep learning-based assessment

PONE-D-21-14181R2

Dear Dr. Kim,

We’re pleased to inform you that your manuscript has been judged scientifically suitable for publication and will be formally accepted for publication once it meets all outstanding technical requirements.

Kind regards,

Niels Bergsland

Academic Editor

PLOS ONE

Additional Editor Comments (optional):

Reviewers' comments:

Reviewer's Responses to Questions

**Comments to the Author**

1. If the authors have adequately addressed your comments raised in a previous round of review and you feel that this manuscript is now acceptable for publication, you may indicate that here to bypass the “Comments to the Author” section, enter your conflict of interest statement in the “Confidential to Editor” section, and submit your "Accept" recommendation.

Reviewer #1: All comments have been addressed

2. Is the manuscript technically sound, and do the data support the conclusions?

Reviewer #1: Yes

3. Has the statistical analysis been performed appropriately and rigorously? 

Reviewer #1: Yes

4. Have the authors made all data underlying the findings in their manuscript fully available?

Reviewer #1: Yes

5. Is the manuscript presented in an intelligible fashion and written in standard English?

Reviewer #1: Yes

6. Review Comments to the Author

Reviewer #1: Thank you for answering the comments. No further comments or questions from me. I wish you good luck.

7. PLOS authors have the option to publish the peer review history of their article (what does this mean?). If published, this will include your full peer review and any attached files.

Reviewer #1: No

---

## [Editor Report · Acceptance letter]

14 Dec 2021

PONE-D-21-14181R2 

Alteration of the corpus callosum in patients with Alzheimer’s disease: Deep learning-based assessment 

Dear Dr. Kim:

I'm pleased to inform you that your manuscript has been deemed suitable for publication in PLOS ONE. Congratulations! Your manuscript is now with our production department. 

Kind regards, 

on behalf of

Dr. Niels Bergsland 

Academic Editor

PLOS ONE